# Cardiac Events Potentially Associated to Remdesivir: An Analysis from the European Spontaneous Adverse Event Reporting System

**DOI:** 10.3390/ph14070611

**Published:** 2021-06-25

**Authors:** Concetta Rafaniello, Carmen Ferrajolo, Maria Giuseppa Sullo, Mario Gaio, Alessia Zinzi, Cristina Scavone, Francesca Gargano, Enrico Coscioni, Francesco Rossi, Annalisa Capuano

**Affiliations:** 1Campania Regional Centre for Pharmacovigilance and Pharmacoepidemiology—Department of Experimental Medicine—Section of Pharmacology “L. Donatelli”, University of Campania “Luigi Vanvitelli”, Via Costantinopoli, 16, 80138 Naples, Italy; carmen.ferrajolo@unicampania.it (C.F.); pina.sullo@libero.it (M.G.S.); mario.gaio@unicampania.it (M.G.); alessia.zinzi@unicampania.it (A.Z.); cristina.scavone@unicampania.it (C.S.); francesco.rossi@unicampania.it (F.R.); annalisa.capuano@unicampania.it (A.C.); 2Department of Anesthesia and Resuscitation, Biomedical Campus University of Rome, 00128 Rome, Italy; f.gargano@unicampus.it; 3AGENAS—Agenzia Nazionale per i Servizi Sanitari Regionali, Via Piemonte 60, 00187 Roma, Italy; coscionienrico@gmail.com

**Keywords:** remdesivir, cardiac events, safety monitoring

## Abstract

Remdesivir was recommended for hospitalized patients with COVID-19. As already reported in the Summary of Product Characteristics, most of remdesivir’s safety concerns are hepatoxicity and nephrotoxicity related. However, some cases have raised concerns regarding the potential cardiac events associated with remdesivir; therefore, the Pharmacovigilance Risk Assessment Committee of the European Medicines Agency requested to investigate all available data. Therefore, we analyzed all Individual Case Safety Reports (ICSRs) collected in the EudraVigilance database focusing on cardiac adverse events. From April to December 2020, 1375 ICSRs related to remdesivir were retrieved from EudraVigilance, of which 863 (62.8%) were related to male and (43.3%) adult patients. A total of 82.2% of all AEs (N = 2604) was serious and one third of the total ICSRs (N = 416, 30.3%) had a fatal outcome. The most frequently reported events referred to hepatic/hepatobiliary disorders (19.4%,), renal and urinary disorders (11.1%) and cardiac events (8.4%). Among 221 cardiac ICSRs, 69 reported fatal outcomes. Other drugs for cardiovascular disorders were reported as suspected/concomitant together with remdesivir in 166 ICSRs (75.1%), 62 of which were fatal. Moreover, the mean time to overall cardiac event was 3.3 days (±2.2). Finally, disproportionality analysis showed a two-fold increased risk of reporting a cardiac adverse event associated with remdesivir compared to both hydroxychloroquine and azithromycin. This study showed that remdesivir could be associated to risk of cardiac events, suggesting a potential safety signal which has not been completely evaluated yet. Further studies are needed to confirm these findings.

## 1. Introduction

Remdesivir, a broad-spectrum antiviral agent, which inhibits viral RNA-dependent RNA polymerase, was the first treatment in Europe recommended with conditional marketing authorization to treat coronavirus disease 2019 (COVID-19) [1]. This recommendation was based on the positive results in terms of time to recovery for hospitalized COVID-19 patients [2]. Meanwhile, the World Health Organization (WHO) released on November 2020 a conditional recommendation against the use of remdesivir in hospitalized patients, regardless of disease severity, since the lack of evidence of survival improvement and other clinical outcomes in these patients. Specifically, this recommendation was clearly reported in the third version (update two) of the living guideline developed by the WHO in collaboration with the non-profit Magic Evidence Ecosystem Foundation (MAGIC) [3]. Moreover, as for all medicines, a risk management plan (RMP) was presented by the marketing holder to ensure rigorous effectiveness and safety monitoring of remdesivir in a real-world setting [4]. According to the RMP, further efficacy and safety data are routinely collected through on-going studies and post-marketing reports and will be regularly reviewed by the Committee for Medicinal Products for Human Use (CHMP) and Pharmacovigilance Risk Assessment Committee (PRAC) of the European Medicines Agency (EMA). In this regard, Fan and collaborators provided a useful and schematic update about the tolerability features related to remdesivir use [5]. In particular, as reported in this manuscript, most of remdesivir’s safety concerns are hepatoxicity and nephrotoxicity related, as already reported in the Summary of Product Characteristics (SmPC) of this antiviral drug [6]. However, evidence about other kinds of potential serious adverse events is still scant. Given the increased use of remdesivir in the context of daily clinical practice, pharmacovigilance activity now takes an extraordinarily important role to promptly characterizing the benefit/risk ratio and improve the clinical outcomes. As a matter of fact, as part of the review of the pandemic summary safety reports for remdesivir, the PRAC already looked at cardiac adverse events (i.e., cases of arrhythmia, hypotension or shock). Additionally, the Italian Medicines Agency (AIFA) identified 11 cases of sinus bradycardia in patients treated with remdesivir. After an evaluation of evidence provided in these cases, the PRAC requested, therefore, to investigate in-depth all available data, including reports from EudraVigilance (EV), clinical trials and the published literature [7]. Therefore, moving in this direction, here we provided an analysis of safety reports collected in the European spontaneous adverse drug reactions (ADR) reporting system focusing on rare adverse events, especially those referred to as cardiac ones.

## 2. Results

### 2.1. Characteristics of Individual Case Safety Reports (ICSRs)

During the study period, 1375 ICSRs with remdesivir as a suspected drug were retrieved from the EudraVigilance ADR report website, of which 863 (62.8%) were related to male patients and 596 (43.3%) to adults (18–64 years of age). Almost all the ICSRs were issued by healthcare professional (N = 1349, 98.1%) and the majority occurred in the non-European Economic Area (non-EEA) (N = 866, 63.0%) (Table 1).

The full resolution of events was reported in one fourth of ICSRs (N = 338), an improvement was seen in 151 cases (11.0%), while events were not resolved yet in 186 patients (13.5%). Almost one third of the total ICSRs (N = 416, 30.3%) had a fatal outcome. The median number of events reported in each ICSR was 2.3 (±2.2), contributing to a total of 3166 adverse events (AEs). A total of 82.2% of all AEs (N = 2604) was reported as serious and almost half of them were included in the “other medically important condition” seriousness criteria (Table 2).

Looking at the overall AEs with a known duration (N = 605 AEs), the mean duration of the outcome was 4.16 (±7.06) days. Moreover, the same analysis, but relating to the 428 AEs “recovered/resolved, with or without sequelae (as reported outcomes)”, the resolution was observed within 5.1 (±8.1) days (data not shown).

Looking at the type of events, the most frequently reported events referred to hepatic/hepatobiliary disorders (19.4%, including “Hepatobiliary disorders” (N = 117) as SOC, and “Hepatobiliary investigations” (N = 496) as HLGT), followed by renal and urinary disorders (11.1%, including “Renal and urinary disorders” (N = 256) as SOC, and “Renal and urinary tract investigations and urinalyses” (N = 95) as HLGT) and cardiac events (8.4%, SOC: “Cardiac disorders” (N = 266)). The most frequently reported AEs are listed in Table 3.

Based on the 890 ICSRs with a known length of therapy, the median duration of remdesivir treatment was 3.0 days (interquartile range—IQR: 2–4). Specifically, the median duration of remdesivir treatment was 2.9 days (interquartile range—IQR: 1–4) for cases in which “drug withdrawal” was reported (N = 353), while 4.0 days (interquartile range—IQR: 3–5) were observed when drug dose was not changed (N = 99).

### 2.2. Cardiac Events

Among 221 ICSRs, with at least one reported cardiac event, the gender difference still remained but was lower (59.7% vs. 40.3%, male vs. female). Sixty-nine ICSRs (31.2%) reported a fatal outcome. However, outcomes with the higher level of resolution were described for 32.5% cardiac ICSRs. Description of cardiac events has been reported in the Appendix A. 

Other medicines for cardiovascular disease (CVD) were reported as suspected/concomitant together with remdesivir in 166 ICSRs (75.1%), 62 of which were fatal. Azithromycin or hydroxychloroquine were reported together with remdesivir in 25.3% of cardiac ICSRs (Table 4).

Looking at the 196 ICSRs with the date of occurrence, the mean time to overall cardiac event was 3.3 days (±2.2). Slight time differences were observed among the type of cardiac events: about 3 days for arrhythmias (N = 162), coronary artery disorders (N = 12), or heart failure, while 5.5 days (±4.9) for cardiac signs and symptoms not elsewhere classified (NEC) (Table 5).

Looking at the probability of reporting ICSRs with a cardiac event, remdesivir was associated with a higher reporting probability of ICSRs with ADRs belonging to the SOC “Cardiac disorders” compared to azithromycin and hydroxychloroquine (ROR 2.1, 95% CI 1.8–2.5, *p* < 0.05; ROR 2.3, 95% CI, 1.9–2.7, *p* < 0.05, respectively).

### 2.3. Description of Other Medicines

Irrespective of the suspected drug remdesivir, other medicines were reported as suspected or concomitant in the ICSRs. Antibacterials for systemic use (N = 1282, 12.2%) were the most frequently reported Anatomical Therapeutic Chemical (ATC) group, followed by antithrombotics (N = 954, 9.1%), systemic corticosteroids (N = 697, 6.7%), analgesics (N = 628, 6.0%), and psycholeptics (N = 530, 5.0%). This trend was consistent across any ICSR and cardiac ICSR (Figure 1).

## 3. Discussion

In this study, we investigated spontaneous reports of AEs related to remdesivir through the analysis of data from the EudraVigilance database providing an overview of the adverse events especially focusing on cardiac effects. During the study period, we analyzed 1375 ICSRs related to mainly male and adult patients for which the healthcare professional was the principal source. Moreover, looking at the reported outcomes, in most of the cases the event was classified as resolved, fully or with sequelae, or is being resolved. Looking at the gender distribution, some studies on ADRs spontaneously reported in pharmacovigilance databases have pointed out that female gender could, in general, represent a risk factor for their occurrence; it could be due to the higher consumption of medicines among women, but also to specific gender-related pharmacokinetic or pharmacodynamic susceptibility and differences in terms of psychosocial, behavioral or cultural factors [8,9]. On the contrary, focusing our analysis on AEs related to remdesivir, our patients were older and mostly of the male gender also when compared to the cases previously reported in a pharmacovigilance study on antiviral-associated adverse events [10]. However, our findings are consistent with demographic characteristics of the enrolled patients in the Adaptive COVID-19 Treatment Trial (ACTT-1) on remdesivir for the treatment of COVID-19 and with data from a post-marketing pharmacovigilance safety study on the WHO global database [1,11]. Notwithstanding remdesivir use, these clinical characteristics are consistent with those of hospitalized COVID-19 patients; in this regard, in fact, regardless the therapeutic strategy, several studies have tried to identify clinical and demographic characteristics, as well as potential factors of worst outcomes, associated to hospitalized COVID-19 patients. Through a retrospective cohort study, Pouw N and collaborators found that a median age of 69 years and the male gender were the prevalent demographic features of hospitalized COVID-19 patients in the Netherlands [12]. Moreover, Gavin W also concluded that an older age and being male may worsen outcomes in COVID-19 [13]. With regard to the gender differences among COVID-19 patients, several studies have already tried to gain sound evidence; specifically, Raimondi F et al. recently published on this topic, but focusing on the hospitalized COVID-19 population. Their findings highlighted that male patients are more prevalent in this setting and that severe COVID-19 could be observed less frequently in women than in men [14]. Looking at the gender distribution among non-hospitalized COVID-19 patients, the available clinical data seem to suggest just the opposite situation; that is, from the laboratory confirmation of SARS-CoV-2 infection to the hospitalization related to COVID-19, the gender distribution does not show significant differences [15]. Moreover, in line with these findings, Grasselli G et al. also demonstrated the prevalence of the male gender (82%) among Italian COVID-19 patients admitted to the intensive care unit [16]. Several factors were claimed in order to explain the gender differences among COVID-19 patients especially those with severe disease; among these, the high expression of angiotensin-converting enzyme 2 in men than in women, sex hormones which could affect the inflammatory response but also the coagulation pattern, preexisting cardiovascular diseases as well as different lifestyle risk factors are those which could explain the observed gender imbalance in patients with COVID-19 [17,18].

The median number of events reported in each ICSRs was 2.3 (±2.2), contributing to a total of 3166 adverse events which were mainly serious (82.2%) and “Other Medically Important Condition” was the most common chosen seriousness criteria. However, we also observed that 30.3% of reported cases resulted in death. Although remdesivir-related adverse events are more serious and more fatal than the other antiviral-related ADRs, however, we cannot exclude the impact of the underlying COVID-19 disease on the outcome of events. As a matter of fact, complications of the physio-pathogenetic mechanism of SARS-CoV-2 infectious disease could be more fatal than any other viral infectious disease requiring antivirals [19]. Looking at the type of events, the mostly reported AEs were related to hepatic/hepatobiliary disorders, renal/urinary tract disorders and cardiac events. In this respect, hepatotoxicity has already been documented in the remdesivir clinical development program, during which some cases have raised concerns regarding potential hepatobiliary disorders associated with its use, both in healthy volunteers and in COVID-19 patients [20]. Moreover, evidence the from post-marketing clinical setting describe several cases of hepatobiliary disorders occurring in patients treated with remdesivir. In this regard, Zampino et al. reported the experience of five patients treated with intravenous remdesivir for COVID-19, as recommended. Findings suggested that remdesivir use could lead to hepatocellular injury with a significant reduction of bilirubin and elevated transaminase levels after three days of antiviral therapy [21]. Furthermore, Montastruc and collaborators performed a pharmacovigilance analysis on VigiBase, the WHO global database of ICSRs, up to June 2020. In line with our results, they found that 34% of total remdesivir reports were suggestive of hepatic adverse events; most of them were related to male patients with a median age of around 55 years. Consistently with our findings, the median duration of remdesivir treatment was around 4 days and most of the analyzed ICSRs were serious and abnormal hepatic enzymes were the most common reported AE. Moreover, disproportionality analysis confirmed this potential association among an increased risk of reporting a hepatic adverse event and remdesivir use [22]. However, both Montastruc and Zampino pointed out that the COVID-19 disease itself could be involved in the occurrence of hepatic damage [21,22]. In a real-world population, nephrotoxicity induced by remdesivir, in terms of a decreased glomerular filtration rate, was found similar to what it has already been observed within trial setting [23]. According to Grein et al.’s study on compassionate use of remdesivir for the treatment of COVID-19, patients reported renal impairments (8%), acute kidney injury (6%) and hematuria (4%) [24]. Moreover, another analysis carried out on VigiBase from February to August 2020 with the aim of verifying the potential correlation between acute renal failure and remdesivir, showed a 20-fold increased risk of reporting this kind of adverse event associated with the antiviral agent compared to lopinavir/ritonavir, tocilizumab and hydroxychloroquine. However, as above, also in this case the authors highlighted that it cannot be ruled out that the role of other factors, such as the COVID-19 disease, older age, poly-therapies and comorbidities, could lead to kidney dysfunction [25]. Meanwhile, the EMA’s CHMP is evaluating a signal for kidney toxicity potentially related to remdesivir, a condition that could have other causes in COVID-19 patients, raised from the Solidarity trial [26]. As a matter of fact, these important potential risks, namely hepatic and renal ones, were considered and clearly reported in the risk management plan of remdesivir [4]. Moreover, as reported in the SmPC, both the liver function and the estimated glomerular filtration rate (eGRF) have to be assessed just before using the antiviral agent and also during the therapy [6].

In addition to the above-described potential toxicities related to remdesivir, evidence about the cardiac adverse events deserve a focus. During the study period, we observed that 221 ICSRs reported cardiac events. Most of them were related to male subjects (59.7%) aged 65–85 years (39.4%). In terms of AEs, this sub-population of ICSRs accounted for a total of 841 cardiac events (median cardiac events per ICSR was two). The most reported cardiac event was cardiac arrhythmias (among this category, bradycardia showed the highest frequency), followed by coronary artery disorders, cardiac and vascular investigations, heart failures, cardiac disorders, (cardiac) signs and symptoms not elsewhere classified (NEC) and myocardial disorders. Cardiac arrhythmia is not described in the remdesivir SmPC; however, in line with our results, a post-marketing study in a real-world setting already suggested that the use of remdesivir could be associated with an increased risk of reporting bradycardia. In particular, the authors found in VigiBase 302 out of 2.603 ICSRs suggestive of a cardiac adverse event related to remdesivir use. Among these cardiac reports, 94 were of bradycardia, most of these were serious and some were fatal (17%) [27].

In the literature, four case reports of sinus bradycardia in patients 26–77 years old with COVID-19 have been described; in particular, this cardiac event occurred within about 3 days after remdesivir initiation and it was fully resolved with the antiviral therapy withdrawal [28,29,30]. The pharmacodynamic mechanism for bradycardia with remdesivir is still unknown; however, some hypotheses have been suggested in this regard. Firstly, remdesivir is an adenosine analog and as such it could act in blocking the atrioventricular node. It is well known that adenosine slows conduction time through the atrioventricular node; therefore, considered the similarity between remdesivir and adenosine triphosphate, theoretically the same effect on the cardiac cells cannot be excluded [31,32]. However, we cannot completely exclude the direct role of COVID-19 in the occurrence of cardiac events. In fact, as well known, COVID-19 could lead to arrhythmias, heart failure and thrombotic complications, even in the absence of a history of cardiovascular diseases [33]. Arrhythmias in COVID-19 may result either primarily due to hypoxia caused from the direct viral tissue involvement of lungs, myocarditis, or an abnormal host immune response, or secondarily because of myocardial ischemia, myocardial strain due to pulmonary hypertension, electrolyte derangements, and intravascular volume imbalances. Arrhythmias are not merely due to the direct effect of COVID-19 infection, but rather are likely as a result of systemic illness [34].

Analyzing the time of occurrence of our cardiac AEs, our findings showed that the mean time to the onset specific for cardiac events was 3.3 days. Touafchia et al. showed in their pharmacovigilance study a slightly shorter median onset of bradycardia (2.4 days; range 1–6) but results with regard to the comparison between remdesivir and hydroxychloroquine are similar to our findings [27]. Our study suggests an increased risk of reporting a cardiac event with remdesivir compared to other well-known cardiotoxic medicines used in the COVID-19 pandemic, such as azithromycin or hydroxychloroquine. In particular, our disproportionality analysis showed a two-fold increased risk of reporting a cardiac adverse event associated with remdesivir compared to both hydroxychloroquine (ROR 2.3, 95% CI, 1.9–2.7, *p* < 0.05) and azithromycin (ROR 2.1, 95% CI 1.8–2.5, *p* < 0.05). With regard to the outcomes for cardiac ICSRs, we observed that 31.2% of these reported a fatal outcome. In this regard, it has been considered, however, that most of the fatal cardiac ICSRs reported, together with remdesivir, other suspected or concomitant drugs indicated for treatment of cardiovascular diseases (90% of fatal cardiovascular ICSR), suggest that these cases were potentially related to patients with a preexisting cardiovascular risk. Moreover, it has also pointed out that these fatal cardiac events are to be referred to hospitalized COVID-19, a clinical condition that itself is predictive of cardiovascular risk with a fatal outcome [35,36,37]. However, our results with other real-world evidence from recently published pharmacovigilance studies, are underlying a potential safety signal for remdesivir, which has not been completely evaluated yet, even if very few cases of cardiac arrest, atrial fibrillation, arrhythmia and supraventricular tachycardia have already been observed in the pre-approval trial [2,10,27]. The risk of cardiac AEs induced by remdesivir remains largely unknown [38].

## 4. Materials and Methods

### 4.1. Data Source

Data on ICSRs were retrieved from the centralized European spontaneous reporting system database (EudraVigilance, EV). The EV, founded by the EMA is a system of collection, management, and analyses of ICSRs of suspected adverse reactions related to medicinal product (ADRs) or following immunization (AEFI). In accordance with EU legislation, the EV database is divided in two modules, such as the Post-Authorization Module (EVPM), to address the collection of ICSRs of suspected ADRs related to medicinal products which are authorized in the European Economic Area (EEA) or the Clinical Trial Module (EVCTM) for collection of ICSRs referring to products which are being studied in clinical trials. The adverse reaction reports collected in the EVPM refer to unsolicited/solicited reports which do not fall under the scope of the Clinical Trials Directive 2001/20/EC reported by a healthcare professional (HCP) or a non-HCP to an EU national competent authority or a marketing authorization holder. After transformation, de-duplication and loading process, public access to data files is provided on suspected the ADR report portal, a website launched by the agency in 2012. In line with the EV access policy, information on patient characteristics (i.e., gender and age-category), adverse events mapped to different levels of MedDRA (Medical Dictionary for Regulatory Activities; meddra.org (accessed on 12 December 2020)) terminology, seriousness, outcomes (e.g., death, hospitalization, etc.), suspected medications, codified by ATC classification, indication of use, dosage, and concomitant/interacting medications can be retrieved as a line listing and/or ICSR form. These data are publicly available for transparency through the EMA website (www.adrreports.eu (accessed on 12 December 2020)) and have been previously described as a valid system for safety surveillance studies [39,40,41].

### 4.2. ICSRs Selection

Remdesivir was the exposure of interest. We selected all ICSRs with remdesivir as a suspected drug reported in EV from 3 April 2020 (date of the first available ICSR in EV related to remdesivir) to 12 December 2020.

### 4.3. Descriptive Analysis

Information on case report characteristics (age, gender, primary source, outcome), number and seriousness criteria of adverse events, number of suspected/concomitant drugs other than remdesivir were provided for all ICSRs. According to the data elements included in the EudraVigilance ADR website reports and possible values, a case is defined as “serious” in accordance with the International Council on Harmonization E2D guidelines or rather when it results in death, is life-threatening, requires hospitalization or prolongs hospitalization, results in persistent or significant disability/incapacity, is a congenital anomaly/birth defect, or results in some other clinically important conditions. Moreover, the outcomes are classified as “Recovered/Resolved”, “Recovering/Resolving”, “Recovered/Resolved With Sequelae”, “Not Recovered/Not Resolved”, “Fatal” and “Unknown” [32]. In case of two or more AEs with different outcome reported in a single ICSRs, the outcome with the lower level of resolution was chosen for classification. ICSRs were classified as fatal if “fatal” was reported as outcome. Time to occurrence of events was calculated as time to onset only for ICSRs that reported both the duration of the therapy and the drug withdrawal as action taken after the occurrence of the event. The frequencies of adverse events were described by Systemic Organ Class (SOC) and High-Level Group Term (HLGT) MedDRA terms. A subgroup analysis was performed for ICSRs with at least one cardiac adverse event, thus belonging to cardiac SOC (named cardiac ICSRs). The other drugs reported as suspected and/or concomitant in addition to remdesivir (reported in ≥1% of ICSRs), were grouped according to the second level of the ATC classification system and were analyzed within overall ICSRs as well as among cardiac ICSRs. To evaluate the amount of ICSRs related to patients with CVD, we evaluated the concomitant presence of cardiovascular agents reported as other suspected or concomitant drugs as proxy of potential CVD.

### 4.4. Disproportionality Analysis

The Reporting Odds Ratio (ROR), with a 95% of Confidence Interval (95% CI) were computed to assess the probability of reporting ICSRs with ADRs belonging to the SOC “Cardiac disorders” for remdesivir compared to other well-known cardiotoxic medicines used in the COVID-19 pandemic, such as azithromycin or hydroxychloroquine. The RORs were computed on ICSRs numbers since these are publicly available on the EMA website. This disproportionality analysis aimed to assess if remdesivir has a lower/higher probability of reporting ICSRs with cardiac events compared with azithromycin and hydroxychloroquine.

### 4.5. Ethical Consideration

Because of data protection regulations, retrieved data included non-identifiable patient information as well as free text narrative from the ICSRs was not available, so no ethics review board approval was required.

## 5. Conclusions

Our findings, in line with other recent studies, highlighted that remdesivir could be potentially associated, compared to other cardiotoxic drugs, to cardiac events, especially cardiac arrhythmias. It is well known that pharmacovigilance studies, if on the one hand, are useful to promptly obtain a suspicion about a potential association between drugs and adverse events pairs, on the other they could not provide evidence on the causal relationship and more or less about a risk between a drug and adverse event. To this end, instead, further confirmatory studies are needed in order to verify a causal association between remdesivir and cardiac events. However, pharmacovigilance studies and disproportionality analyses are deemed necessary in providing real-world data essential to better and early identifying of new drug safety issues, especially within a global public health emergency. Therefore, waiting for results from further studies, both clinicians and regulators should consider the potential cardiac events to remdesivir.

## Figures and Tables

**Figure 1 pharmaceuticals-14-00611-f001:**
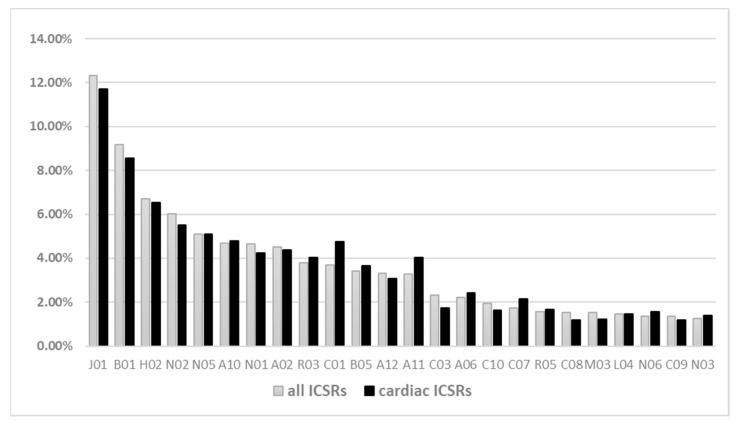
Distribution (not mutually exclusive) of other suspected/concomitant drugs reported in the remdesivir ICSRs. Legend: J01: antibacterials for systemic use; B01: antithrombotic agents; H02: corticosteroids for systemic use; N02: analgesics; N05: psycholeptics; A10: drugs used in diabetes; N01: anesthetics; A02: drugs for acid related disorders; R03: drugs for obstructive airway diseases; C01: cardiac therapy; B05: blood substitutes and perfusion solutions; A12: mineral supplements; A11: vitamins; C03: diuretics; A06: drugs for constipation; C10: lipid-modifying agents; C07: beta-blocking agents; R05: cough and cold preparations; C08: calcium channel blockers; M03: muscle relaxants; L04: immunosuppressants; N06: psychoanaleptics; C09: agents acting on the renin–angiotensin system; N03: antiepileptics.

**Table 1 pharmaceuticals-14-00611-t001:** Demographic characteristics of Individual Case Safety Reports (ICSRs) involving remdesivir reported in the EudraVigilance spontaneous reporting system from 3 April 2020 to 12 December 2020.

	Individual Case Safety Reports 1375 (%)
Gender	
Male	863 (62.8)
Female	477 (34.7)
Not Specified	35 (2.5)
Age group	
Paediatrics (>18 Years)	12 (0.9)
Adult (18–64 Years)	596 (43.3)
Elderly (65–85 Years)	503 (36.6)
Very Elderly (>85 Years)	93 (6.8)
Not Specified	171 (12.4)
Type of reporter	
Health Care Professional	1349 (98.1)
Non-Health Care Professional	26 (1.9)
Country	
European Economic Area	509 (37.0)
Non-European Economic Area	866 (63.0)
Outcome ^1^	
Fatal	416 (30.3)
Recovered/Resolved	338 (24.6)
Not Recovered/Not Resolved	186 (13.5)
Recovering/Resolving	151 (11.0)
Recovered/Resolved With Sequelae	5 (0.4)
Unknown	279 (20.3)
Adverse Events	
Total Number	3166
Mean AE per ICSR (±SD)	2.3 (±2.24)

^1^ In case of more than one event/outcome, the worst outcome was considered.

**Table 2 pharmaceuticals-14-00611-t002:** Characteristics of adverse events.

	Number of Events 3166 (%)
**Seriousness**	
Not Serious	562 (17.8)
Seriousness Criteria, of which	2604 (82.2)
Other Medically Important Condition	1293 (49.7)
Results in Death	703 (27.0)
Caused/Prolonged Hospitalisation	398 (15.3)
Life Threatening	179 (6.9)
Disabling	29 (1.1)
Congenital Anomaly	2 (0.1)
**Outcome**	
Fatal	703 (22.2)
Not Recovered/Not Resolved	577 (18.2)
Recovered/Resolved	644 (20.3)
Recovering/Resolving	293 (9.3)
Recovered/Resolved With Sequelae	6 (0.2)
Unknown	943 (29.8)
Adverse Event Duration, Days, Mean (±SD)	4.16 (7.06)

**Table 3 pharmaceuticals-14-00611-t003:** Top 10 distribution of adverse events by MedDRA System Organ Class, and top three High-Level Group Terms (HLGT) within each System Organ Class (SOC).

Adverse Events by MedDRA SOC and HLGT	Adverse Events 3166 (%)
Investigations	792 (25.0)
Hepatobiliary investigations	496 (15.7)
Renal and urinary tract investigations and urinalyses	95 (3.0)
Haematology investigations (incl. blood groups)	54 (1.7)
Other	147 (4.6)
General disorders and administration site conditions	413 (13.0)
Fatal outcomes	200 (6.3)
General system disorders NEC ^1^	140 (4.4)
Therapeutic and nontherapeutic effects (excluding toxicity)	24 (0.8)
Other	49 (1.5)
Respiratory, thoracic and mediastinal disorders	286 (9.0)
Respiratory disorders NEC ^1^	195 (6.2)
Lower respiratory tract disorders (excluding obstruction and infection)	38 (1.2)
Infections—pathogen unspecified	14 (0.4)
Other	39 (1.2)
Cardiac disorders	266 (8.4)
Cardiac arrhythmias	230 (7.3)
Coronary artery disorders	16 (0.5)
Heart failures	10 (0.3)
Other	10 (0.3)
Renal and urinary disorders	256 (8.1)
Renal disorders (excluding nephropathies)	244 (7.7)
Urinary tract signs and symptoms	6 (0.2)
Nephropathies	4 (0.1)
Other	2 (0.1)
Infections and infestations	249 (7.9)
Viral infectious disorders	147 (4.6)
Infections—pathogen unspecified	63 (2.0)
Bacterial infectious disorders	30 (0.9)
Other	9 (0.3)
Hepatobiliary disorders	117 (3.7)
Hepatic and hepatobiliary disorders	115 (3.6)
Other	2 (0.1)
Vascular disorders	116 (3.7)
Decreased and nonspecific blood pressure disorders and shock	87 (2.7)
Embolism and thrombosis	11 (0.3)
Vascular hypertensive disorders	8 (0.3)
Other	10 (0.3)
Nervous system disorders	110 (3.5)
Neurological disorders NEC ^1^	32 (1.0)
Central nervous system vascular disorders	21 (0.7)
Seizures (incl. subtypes)	13 (0.4)
Other	44 (1.4)
Skin and subcutaneous tissue disorders	101 (3.2)
Epidermal and dermal conditions	68 (2.1)
Skin appendage conditions	13 (0.4)
Angioedema and urticaria	10 (0.3)
Other	10 (0.3)
Other SOCs	920 (29.1)

^1^ “Not elsewhere classified” (NEC) is a standard abbreviation used to denote groupings of miscellaneous terms that do not readily fit into other hierarchical classifications within a particular SOC.

**Table 4 pharmaceuticals-14-00611-t004:** Characteristics of 221 cardiac Individual Case Safety Reports.

	Cardiac Individual Case Safety Reports 221 (%)
Gender	
Female	89 (40.3)
Male	132 (59.7)
Age group	
12–17 Years	1 (0.5)
18–64 Years	83 (37.6)
2 Months–2 Years	1 (0.5)
65–85 Years	87 (39.4)
More Than 85 Years	31 (14.0)
Not Specified	18 (8.1)
Outcome	
Fatal	69 (31.2)
Not Recovered/Not Resolved	35 (15.8)
Recovered/Resolved	60 (27.1)
Recovered/Resolved With Sequelae	3 (1.4)
Recovering/Resolving	9 (4.1)
Unknown	45 (20.4)
N. of cardiac events per ICSR	
1	178 (80.5)
2	28 (12.7)
3	9 (4.1)
4	3 (1.4)
5	3 (1.4)
Overall Adverse Events	
Total number	841
Median per ICSR	2 (1–17)
Other drugs	
Total number	2372
Median per ICSR	8 (0–52)
Other drugs for cardiovascular diseases	
Yes	166 (75.1)
No	55 (24.9)
Other cardiotoxic drugs (azithromycin or hydroxychloroquine)	
Yes	56 (25.3)
No	165 (74.7)

**Table 5 pharmaceuticals-14-00611-t005:** Description of time-to-onset for cardiac events.

	No. of Events	Mean of Time to Onset (Days)	StdDev
Cardiac Events with time information	196	3.3	±2.2
Cardiac arrhythmias	162	3.3	±2.3
Coronary artery disorders	12	3.3	±2.1
Cardiac and vascular investigations (ex. enzyme tests)	11	2.6	±1.4
Heart failures	7	3.1	±1.7
Cardiac disorders, signs and symptoms NEC ^1^	2	5.5	±4.9
Myocardial disorders	2	2.5	±2.1

^1^ “Not elsewhere classified” (NEC) is a standard abbreviation used to denote groupings of miscellaneous terms that do not readily fit into other hierarchical classifications within a particular SOC.

## Data Availability

Authors can confirm that all relevant data are included in the article.

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
