# Peer review of "Cardiac Events Potentially Associated to Remdesivir: An Analysis from the European Spontaneous Adverse Event Reporting System"

_pharmaceuticals, 2021, doi:10.3390/ph14070611_

Round 1
Reviewer 1 Report
COVID itself is associated with causing cardiac issues (myocarditis/pericarditis) - how do we know if the reported events are truly 2/2 remdesivir vs. the disease itself (COVID-19 as a Possible Cause of Myocarditis and Pericarditis - American College of Cardiology (acc.org)? This should be listed/discussed as a limitation in the discussion.
Author Response
1) COVID itself is associated with causing cardiac issues (myocarditis/pericarditis) - how do we know if the reported events are truly 2/2 remdesivir vs. the disease itself (COVID-19 as a Possible Cause of Myocarditis and Pericarditis - American College of Cardiology (acc.org)? This should be listed/discussed as a limitation in the discussion.
Answer
Dear Reviewer, thank you for your comments which give us the opportunity to better explain some key points inherent to the pharmacovigilance science before answering this question. First, ‘adverse event’ and ‘adverse drug reaction’ terms are often used as synonyms, however, the first one is defined as ‘Medical occurrence temporally associated with the use of a medicinal product, but not necessarily causally related’ and the second one could be defined as ‘a response to a medicinal product which is noxious and unintended’; therefore, the main difference between an adverse event and an adverse drug reaction is that a causal relationship is suspected for the latter. Pharmacovigilance databases, both at National and International level, have been created to promote the collection and then the analysis of adverse events (or suspected adverse reactions) for drugs to create meaningful safety information. Signal detection from individual case reports (a document providing information related to an individual case of a suspected adverse reaction due to a medicine) is the most important aspect in pharmacovigilance science. Specifically, a safety signal is defined as ‘information on a new or known adverse event that is potentially caused by a medicine and that warrants further investigation” suggesting that it is essentially a hypothesis of correlation between drug/adverse event pair which has to be confirmed through ad-hoc studies. Therefore, the evidence in a signal is not conclusive and is only an early indication (preliminary), as it may change substantially over time as more data accumulates. Disproportionality analysis is primarily a tool to generate hypotheses on possible relations between drugs and adverse events. It is based on the statistically discrepancy between observed and expected numbers of reports, for any given combination of drug and adverse event. Moreover, disproportionality analysis is generally recommended and necessary for large databases. Thus, considering the above premises and taking in to account your question, we cannot rule out a direct role of COVID-19 itself in the occurrence of cardiac complications especially in hospitalized patients as well as we cannot rule at the same time the potential impact of remdesivir exposure on these events. In fact, looking at the results of the disproportionality analysis that we made (using the reporting odds ratio as method for), also for azithromycin and hydroxychloroquine individual case safety report, the underlying Covid-19 would have been a potential factor for cardiac events occurrence; nevertheless, the results suggested a higher “disproportionate” rate of reporting cardiac adverse events for remdesivir compared both with azithromycin and hydroxychloroquine. On the other side, we cannot overlook the available evidence which have suggested that the Covid-19 could lead to myocarditis and pericarditis. Definitively, pharmacovigilance database descriptive analysis doesn’t provide information about the potential role of an underlying disease, in our case the Covid-19, on the occurrence of an adverse event, such as the cardiac one. In order to quantifying the potential impact of Covid-19 vs remdesivir on cardiac complications, would have been suitable an etiological study. However, pharmacovigilance studies, if on one hand are useful to promptly give a suspicion about a potential association between drugs/adverse events pairs, on the other they could not provide evidence on causal relationship and more less about a risk between a drug and adverse event. Definitely, we have stressed more this weakness, in the discussion section, see lines 471 - 479.
However, in our study, we found only one case reporting myocarditis and no case of pericarditis.

Reviewer 2 Report
Editorial comments:
The efficacy of Remdesivir itself is debatable and so if the adverse effect profile is well studied, then it might drastically affect the risk/benefit ratio and perhaps making it easier to make a decision on its use. We may use Remdesivir in future for management of some other disease as well. So the topic of the study does appear to have a good impact.
Here are the concerns which can be better addressed.
1) It will be informative to know, what exactly constituted “Cardiac adverse effect” and how was it defined. For eg. When was it considered bradycardia or how was the MI defined? This will give a better insight into how accurate/reliable the diagnosis was and it will also point in the direction of the mechanism .
2) COVID -19 itself tends to cause myocarditis, thromboembolism, myocardial infarctions and arrhythmias including tachyarrhythmias and bradycarrhythmias. Though these AE have been reported by physicians, is it possible to completely rule out the effect of COVID in these pts and it not being Remdesivir?
3)Also most of the patients were already on cardiac meds making it a possibility that the underlying cardiac disease was just unmasked by the stress of the events going on and not necessarily Remdesevir causing it. We need a control group to really rule out the confounding factors. This is addressed in the discussion part, but would prefer it to be elaborated on.
4)For statement regarding hydroxychloroquine and azithromycin, do we know how many patients received HCQ+ AZT in the database. Also as per the database, how many pt received Remdesivir (1375 of whom had ICSRs). We need to know this especially looking at mortality being so high in the study group.
5)For recommending a safety update we will need in-depth analysis to confirm that there was at least clear association (if not causation). There are too many confounding factors.
Author Response
Reviewer 2
Editorial comments:
The efficacy of Remdesivir itself is debatable and so if the adverse effect profile is well studied, then it might drastically affect the risk/benefit ratio and perhaps making it easier to make a decision on its use. We may use Remdesivir in future for management of some other disease as well. So the topic of the study does appear to have a good impact.
Here are the concerns which can be better addressed.
1) It will be informative to know, what exactly constituted “Cardiac adverse effect” and how was it defined. For eg. When was it considered bradycardia or how was the MI defined? This will give a better insight into how accurate/reliable the diagnosis was and it will also point in the direction of the mechanism.
Answer:
We thank the reviewer for this opportunity to clarify. We defined “cardiac adverse events” according to the System Organ Class (SOC) of Medical Dictionary for Regulatory Activities (MedDRA) Terminology. MedDRA is an international hierarchical system of medical terminology developed by the International Council for Harmonisation of Technical Requirements for Pharmaceuticals for Human Use (ICH) with the aim to standardize the international medical terminology for regulatory communication. “The objective was to agree on a unified medical terminology for regulatory activities that overcomes the limitations of current terminologies, is internationally accepted, and is supported by long-term maintenance” [Introduction Guide MedDRA 24.0. Available at: https://admin.new.meddra.org/sites/default/files/guidance/file/intguide_%2024_0_English.pdf ]. This tool is particularly used in pharmacovigilance field in order to categorize the events according to the involved anatomical organ/tissue.
A SOC is the highest level of the hierarchy that provides the broadest concept for data retrieval. “Cardiac disorders” includes any medical event selected partly on an anatomic basis (endocardial, myocardial and pericardial disorders, coronary artery disorders, and valve disorders) and partly by pathophysiology (neoplasms, arrhythmias, cardiac failure, congenital cardiac disorders, and cardiac signs and symptoms). Therefore, no clinical criteria have been used for cardiac adverse events definition as we have already reported in the ‘Method’ section: “A subgroup analysis was performed for ICSRs with at least one cardiac adverse event, thus belonging to cardiac SOC (named cardiac ICSRs).”
Moreover, to give more details about reported cardiac events we have also provided a description of any reported cardiac event in the supplementary table.
2) COVID -19 itself tends to cause myocarditis, thromboembolism, myocardial infarctions and arrhythmias including tachyarrhythmias and bradycarrhythmias. Though these AE have been reported by physicians, is it possible to completely rule out the effect of COVID in these pts and it not being Remdesivir?
Answer:
Dear Reviewer, thank you for your comments which give us the opportunity to better explain some key points inherent to the pharmacovigilance science before answering this question. First, ‘adverse event’ and ‘adverse drug reaction’ terms are often used as synonyms, however, the first one is defined as ‘Medical occurrence temporally associated with the use of a medicinal product, but not necessarily causally related’ and the second one could be defined as ‘a response to a medicinal product which is noxious and unintended’; therefore, the main difference between an adverse event and an adverse drug reaction is that a causal relationship is suspected for the latter. Pharmacovigilance databases, both at National and International level, have been created to promote the collection and then the analysis of adverse events (or suspected adverse reactions) for drugs to create meaningful safety information. Signal detection from individual case reports (a document providing information related to an individual case of a suspected adverse reaction due to a medicine) is the most important aspect in pharmacovigilance science. Specifically, a safety signal is defined as ‘information on a new or known adverse event that is potentially caused by a medicine and that warrants further investigation” suggesting that it is essentially a hypothesis of correlation between drug/adverse event pair which has to be confirmed through ad-hoc studies. Therefore, the evidence in a signal is not conclusive and is only an early indication (preliminary), as it may change substantially over time as more data accumulates. Disproportionality analysis is primarily a tool to generate hypotheses on possible relations between drugs and adverse events. It is based on the statistically discrepancy between observed and expected numbers of reports, for any given combination of drug and adverse event. Moreover, disproportionality analysis is generally recommended and necessary for large databases. Thus, considering the above premises and taking in to account your question, we cannot rule out a direct role of COVID-19 itself in the occurrence of cardiac complications especially in hospitalized patients as well as we cannot rule at the same time the potential impact of remdesivir exposure on these events. In fact, looking at the results of the disproportionality analysis that we made (using the reporting odds ratio as method for), also for azithromycin and hydroxychloroquine individual case safety report, the underlying Covid-19 would have been a potential factor for cardiac events occurrence; nevertheless, the results suggested a higher “disproportionate” rate of reporting cardiac adverse events for remdesivir compared both with azithromycin and hydroxychloroquine. On the other side, we cannot overlook the available evidence which have suggested that the Covid-19 could lead to arrhythmias, heart failure and thrombotic complications. Definitively, pharmacovigilance database descriptive analysis doesn’t provide information about the potential role of an underlying disease, in our case the Covid-19, on the occurrence of an adverse event, such as the cardiac one. In order to quantifying the potential impact of Covid-19 vs remdesivir on cardiac complications, would have been suitable an etiological study. However, pharmacovigilance studies, if on one hand are useful to promptly give a suspicion about a potential association between drugs/adverse events pairs, on the other they could not provide evidence on causal relationship and more less about a risk between a drug and adverse event. However, we have stressed more this weakness, in the discussion section, see lines 471 - 479.
3)Also most of the patients were already on cardiac meds making it a possibility that the underlying cardiac disease was just unmasked by the stress of the events going on and not necessarily Remdesevir causing it. We need a control group to really rule out the confounding factors. This is addressed in the discussion part, but would prefer it to be elaborated on.
Answer:
Dear Reviewer, thank you for your suggestion; we agree with you about the need of a control group, but as explained above, the aim of pharmacovigilance studies is to generate hypothesis on potential association between drug/event pair that need to be further evaluated through formal pharmacoepidemiological studies as recommended by the EU Good Pharmacovigilance Practices (GVP). This is even true when pharmacovigilance database (Eudravigilance in our study) is analyzed through the on-line access tool through which all spontaneous reports as aggregated data are the only accessible information. Moreover, since information about the exposure time to drugs, both suspected and concomitant ones, could be not available, we cannot ascertain if cardiac medicines have been used as chronic or as immediate medical care. Indeed, among the cardiac ICSRs, both fatal and not-fatal, most of the cardiac concomitant drugs belonged to the ATC group C01 which comprises active ingredients both for acute (e.g. dobutamine, dopamine) or chronic conditions (e.g. antiarrhythmic and nitro drugs).
4)For statement regarding hydroxychloroquine and azithromycin, do we know how many patients received HCQ+ AZT in the database. Also as per the database, how many pt received Remdesivir (1375 of whom had ICSRs). We need to know this especially looking at mortality being so high in the study group.
Answer:
We used the reports involving HCQ or AZT as “comparator” by assuming that these two medicines have been utilized in COVID patients in the same period of remdesivir, since no other drugs were authorised for COVID treatment. Moreover, HCQ and AZT are well-known associated to cardiotoxicity, so that, good comparator for this disproportionality analysis.
Moreover, according to the source of the data, this database collects suspected adverse drug reactions reporting data, not drug exposure data. So, these databases do not provide the number of patients assuming medicines. Moreover, given that we don’t know data on drug exposure, we were able to only describe events with reported fatal outcome but not mortality rate.
5)For recommending a safety update we will need in-depth analysis to confirm that there was at least clear association (if not causation). There are too many confounding factors.
Answer:
Dear Reviewer, thank you for your precious comment. We have modified the conclusions explaining better our results and stressing the need of further confirmatory studies which could verify a clear causal association between remdesivir and cardiac events.
“Our findings, in line with other recent studies, highlighted that remdesivir could be potentially associated, compared to other cardiotoxic drugs, to cardiac events, especially cardiac arrhythmias. It is well known that pharmacovigilance studies, if on one hand are useful to promptly give a suspicion about a potential association between drugs/adverse events pairs, on the other they could not provide evidence on causal relationship and more less about a risk between a drug and adverse event. To this end, instead, further confirmatory studies are needed in order to verify a causal association between remdesivir and cardiac events. However, pharmacovigilance studies and disproportionality analyses are deemed necessary to providing real-world data essential to better and early identifying new drug safety issue, especially within a global public health emergency. Therefore, waiting for results from further studies, both clinicians and regulators should consider the potential cardiac events to remdesivir”.

Reviewer 3 Report
The manuscript entitled "cardiac events related to remdesivir use: is there a need of a new safety update?" is interesting and highlights an important topic that is worth further investigations especially with remedisivr being routinely used in the treatment of COVID-19 hospitalized patients. However I have several concerns that needs to be addressed.
- The title needs modification as it currently sounds more like a review article
- My major concern is that I struggled to find the correlation between remdesivir use and cardiac events, the majority of patients were receiving drugs for cardiovascular diseases concomitantly with remdesivir and it is not clear whether cardiac events were related to remdesivir or due to patient's conditions and diseases. Moreover, comparing remdesivir with hydroxychloroquine + azithromycin is not clearly described as I could not find the direct comparison between patients receiving remd. or the other meds. More importantly the 2 fold increase in cardiac events with remd. compared to the other drugs is not clearly described as no information provided regarding the other medications, the prescribing pattern and the number of patients in each group.
- Why only limited to hydroxychloroquine and azithromycin? especially that the current recommendations is against using those medications in routine practice. So my expectations is that the prescribing pattern of those meds was reduced during the later months of 2020 as compared to the beginning of the year. I understand the cardiotoxic effects of those medications but still several other options currently used can be added or at least discussed.
- Providing more details regarding the type of cardiac events (QT prolongations, type of arrhythmia etc..) can be very useful.
- The manuscript have several typos, grammar mistakes, spelling mistakes.
- Several acronyms used needs to be spelled out in first appearance also try to be minimizing the capitalization of letters as this is a bit confusing for the reader with the large number of acronyms used.
- The units needs to be added when discussing the duration of treatment or any other measures. Several numbers are described without units.
- Double check the spelling of the medications mentioned in the manuscript
- Line 44 the sentence is not clear (...as first?. its?...)
- Table 1 title is not clear
- Table 3 the word excl. is not described
- Line 121 the sentence is also not easy to understand
- Paragraph 2.4 more details needed this paragraph is very poor.
- Table for the Percent title (%) is missing
- Overall the manuscript provides details about adverse events with remdesivir and also highlights some gender differences but the cardiac events and the relation between remdesivir and those events is not clearly described.
Author Response
Reviewer 3
The manuscript entitled "cardiac events related to remdesivir use: is there a need of a new safety update?" is interesting and highlights an important topic that is worth further investigations especially with remedisivr being routinely used in the treatment of COVID-19 hospitalized patients. However I have several concerns that needs to be addressed.
1) The title needs modification as it currently sounds more like a review article
Answer
The title has been modified as follows: Cardiac events potentially associated to remdesivir use: an analysis from European spontaneous reporting system
2) My major concern is that I struggled to find the correlation between remdesivir use and cardiac events, the majority of patients were receiving drugs for cardiovascular diseases concomitantly with remdesivir and it is not clear whether cardiac events were related to remdesivir or due to patient's conditions and diseases. Moreover, comparing remdesivir with hydroxychloroquine + azithromycin is not clearly described as I could not find the direct comparison between patients receiving remd. or the other meds. More importantly the 2 fold increase in cardiac events with remd. compared to the other drugs is not clearly described as no information provided regarding the other medications, the prescribing pattern and the number of patients in each group.
Answer
Dear Reviewer,
Thank you for your precious comment that gives also us the possibility to provide here more details which could be useful to clarify your concerns. First, ‘adverse event’ and ‘adverse drug reaction’ terms are often used as synonyms, however, the first one is defined as ‘Medical occurrence temporally associated with the use of a medicinal product, but not necessarily causally related’ and the second one could be defined as ‘a response to a medicinal product which is noxious and unintended’; therefore, the main difference between an adverse event and an adverse drug reaction is that a causal relationship is suspected for the latter. Pharmacovigilance databases, both at National and International level, have been created to promote the collection and then the analysis of adverse events (or suspected adverse reactions) for drugs to create meaningful safety information. Signal detection from individual case reports (a document providing information related to an individual case of a suspected adverse reaction due to a medicine) is the most important aspect in pharmacovigilance science. Specifically, a safety signal is defined as ‘information on a new or known adverse event that is potentially caused by a medicine and that warrants further investigation” suggesting that it is essentially a hypothesis of correlation between drug/adverse event pair which has to be confirmed through ad-hoc studies. Therefore, the evidence in a signal is not conclusive and is only an early indication (preliminary), as it may change substantially over time as more data accumulates. Disproportionality analysis is primarily a tool to generate hypotheses on possible relations between drugs and adverse events. It is based on the statistically discrepancy between observed and expected numbers of reports, for any given combination of drug and adverse event. Moreover, disproportionality analysis is generally recommended and necessary for large databases. Thus, considering the above premises and taking in to account your question we cannot rule out that cardiac concomitant drugs, which in turn could suggest a cardiac disease as a pre-existing pathological condition, have could be played a role on the cardiac events occurrence. But, at the same time we cannot exclude potential impact of remdesivir exposure on these events. In fact, looking at the results of the disproportionality analysis that we made (using the reporting odds ratio as method for), also for azithromycin and hydroxychloroquine individual case safety report, the utilization of cardiac medicines would have been a potential factor for cardiac events occurrence; nevertheless, the results suggested a higher “disproportionate” rate of reporting cardiac adverse events for remdesivir compared both with azithromycin and hydroxychloroquine. More in depth, regarding the disproportionality analysis that we made using the reporting odds ratio (ROR), we compared the ICSRs with adverse event belonging to the system organ class (SOC) ‘cardiac disorders’ between remdesivir and azithromycin and remdesivir and hydroxychloroquine. In this regard, here we provide you the two-by-two contingency table used for the above analysis:
1) Remdesivir vs azithromycin
|
|
N. of ICSRs with adverse event belonging to the SOC cardiac events
|
N. of ICSRs with adverse event belonging to the other SOC excluding the cardiac SOC
|
|
REMDESIVIR |
197 |
1134 |
|
AZITHROMYCIN |
1029 |
12522 |
ROR= 2,11 (CI95% 1,8 – 2,5) z-statistic=8,9 p<0,0001
2) Remdesivir vs hydroxychloroquine
|
|
N. of ICSRs with adverse event belonging to the SOC cardiac events
|
N. of ICSRs with adverse event belonging to the other SOC excluding the cardiac SOC
|
|
REMDESIVIR |
197 |
1134 |
|
HYDROXYCHLOROQUINE |
1152 |
14943 |
ROR= 2.2534 (CI95% 1.9 - 2.7) z-statistic= 9.8 p<0,0001
Through this disproportionality analysis we found that the remdesvir had significantly higher ROR for cardiac SOC compared both to azithromycin or hydroxychloroquine, however, these results should be interpreted like a suspicion of association between remdesivir and cardiac events rather than a cardiac risk for this drug. As you can see, this disproportionality analysis doesn’t take into account other medications, prescribing pattern and number of patients in each group.
3) Why only limited to hydroxychloroquine and azithromycin? especially that the current recommendations is against using those medications in routine practice. So my expectations is that the prescribing pattern of those meds was reduced during the later months of 2020 as compared to the beginning of the year. I understand the cardiotoxic effects of those medications but still several other options currently used can be added or at least discussed.
Answer
We choose the reports involving HCQ or AZT as “comparator” based on the following assumptions: 1) the these two medicines have been utilized in COVID patients in the same period of remdesivir; 2) HCQ and
AZT are well-known to be associated to cardiotoxicity. So that, they represent a good comparator for this disproportionality analysis.
Moreover, according to the source of the data, this database collects suspected adverse drug reactions reporting data, not drug exposure data. So, these databases do not provide prescribing pattern.
4) Providing more details regarding the type of cardiac events (QT prolongations, type of arrhythmia etc..) can be very useful.
Answer
We thank the reviewer for the opportunity to clarify.
A description of 288 cardiac events referring to 221 ICSR has been provided in a supplementary table.
Supplementary Table: Description of type of cardiac events by preferred terms MedDRA
5)The manuscript have several typos, grammar mistakes, spelling mistakes.
Amended as requested.
6) Several acronyms used needs to be spelled out in first appearance also try to be minimizing the capitalization of letters as this is a bit confusing for the reader with the large number of acronyms used.
Amended as requested.
7) The units needs to be added when discussing the duration of treatment or any other measures. Several numbers are described without units.
Amended as requested.
8) Double check the spelling of the medications mentioned in the manuscript
Amended as requested.
9) Line 44 the sentence is not clear (...as first?. its?...)
Amended as requested.
10) Table 1 title is not clear
Amended as requested.
11) Table 3 the word excl. is not described
Amended as requested.
12) Line 121 the sentence is also not easy to understand
Amended as requested.
13) Paragraph 2.4 more details needed this paragraph is very poor.
The paragraph 2.4 has been deleted and the disproportionality analysis has moved at the end of paragraph 2.2.
14) Table for the Percent title (%) is missing
Amended as requested.
15) Overall the manuscript provides details about adverse events with remdesivir and also highlights some gender differences but the cardiac events and the relation between remdesivir and those events is not clearly described.
Answer:
According to the type of analysis, the aim of this pharmacovigilance study is to describe the safety profile of real world by using spontaneous reporting data and to generate a hypothesis about a potential association between remdesivir and cardiac events. This hypothesis needs to be validated by formal pharmacoepidemiology studies.

Round 2
Reviewer 1 Report
n/a
Reviewer 2 Report
The findings of the study needs to be published so as to share the possible implications of indiscriminate use of Remdesivir, with the medical community.